# The Development of a Personalized Symptom Management Mobile Health Application for Persons Living with HIV in China

**DOI:** 10.3390/jpm11050346

**Published:** 2021-04-25

**Authors:** Shuyu Han, Yaolin Pei, Lina Wang, Yan Hu, Xiang Qi, Rui Zhao, Lin Zhang, Wenxiu Sun, Zheng Zhu, Bei Wu

**Affiliations:** 1School of Nursing, Peking University, Beijing 100191, China; 18111170001@fudan.edu.cn; 2Rory Meyers College of Nursing, New York University, New York, NY 10010, USA; yp22@nyu.edu (Y.P.); xq450@nyu.edu (X.Q.); 3School of Medicine, Huzhou University, Huzhou Central Hospital, Huzhou 313000, China; 13587278357@163.com; 4School of Nursing, Fudan University, Shanghai 200032, China; zhruicc@foxmail.com (R.Z.); zhengzhu@fudan.edu.cn (Z.Z.); 5Fudan University Centre for Evidence-based Nursing: A Joanna Briggs Institute Centre of Excellence, Shanghai 200032, China; 6Children’s Hospital of Fudan University, Shanghai 201102, China; 7Shanghai Public Health Clinical Center Affiliated with Fudan University, Shanghai 201508, China; zhanglin@shphc.org.cn (L.Z.); sunwenxiu@shphc.org.cn (W.S.)

**Keywords:** HIV, mobile health, smartphone application, symptom management

## Abstract

Persons living with HIV (PLWH) continuously experience symptom burdens. Their symptom prevalence and severity are also quite different. Mobile health (mHealth) applications (apps) offer exceptional opportunities for using personalized interventions when and where PLWH are needed. This study aimed to demonstrate the development process of the symptom management (SM) app and the structure and content of it. Our research team systematically searched for evidence-based resources and summarized up-to-date evidence for symptom management and health education. Our multidisciplinary research team that included physicians, nurses, software engineers, and nursing professors, evaluated the structure and content of the drafted app. Both quantitative data and qualitative results were collected at a group discussion meeting. Quantitative data were scores of sufficient evidence, situational suitability, practicability, cost-effectiveness, and understandability (ranged from one to four) for 119 items of the app contents, including the health tracking module, the self-assessment module, coping strategies for 18 symptoms (80 items), medication management, complementary therapy, diet management, exercise, relaxation techniques, and the obtaining support module. The SM app was comprised of eight modules and provided several personalized symptom management functions, including assessing symptoms and receiving different symptom management strategies, tracking health indicators, and communicating with medical staff. The SM app was a promising and flexible tool for HIV symptom management. It provided PLWH with personalized symptom management strategies and facilitated the case management for medical staff. Future studies are needed to further test the app’s usability among PLWH users and its effects on symptom management.

## 1. Introduction

Since the development of antiretroviral therapy (ART), persons living with HIV (PLWH) can achieve a relatively satisfying life expectancy [1]. However, symptoms and related health outcomes have brought significant challenges and distress to the quality of life for PLWH. Many factors such as HIV infection, opportunistic infections and comorbidities, ART toxicity, and social discrimination contribute to the onset and progression of symptoms among PLWH [2,3]. Although ART helps to decrease symptom intensity [4], it also brings on additional symptoms such as lipodystrophy, insomnia, and rash [3]. Evidence showed that both the number and frequency of symptoms were not associated with CD4 count levels, which indicated that those symptoms persisted throughout the HIV infection trajectory in PLWH [4]. PLWH usually suffer from physical and psychological symptoms simultaneously [5]. Common symptoms of PLWH include fatigue, depression, pain, rash, and insomnia. The average symptom counts can be as high as 8–17 [6,7,8]. Previous studies reported a different but high prevalence of these common symptoms, with a range of 43–78% [6,7,8]. PLWH are burdened with multiple symptoms. They also have different symptoms of distress and unmet needs further exacerbate their situations. Therefore, the long-term and complexity of HIV symptoms call for more personalized and sustainable symptoms management strategies.

Chinese PLWH receive free ART based on the National Strategies of Four Frees and One Care for HIV/AIDS [9]. However, only publicly-funded domestic antiretroviral medications are free of charge. The integrase strand transfer inhibitors (INSTIs), which are the internationally recommended first-line treatment medication for PLWH [10], by contrast, are imported and needed to be paid out-of-pocket. Therefore, due to the issue of affordability, instead of taking these imported INSTIs, most Chinese PLWH choose to take free ART medications, e.g., efavirenz (EFV) or lopinavir/ritonavir (LPV/r). These medications may increase their risks for medication toxicity and symptoms of distress (EFV may lead to rash and sleep disorders, LPV/r may lead to diarrhea [11]). Evidence also showed that PLWH in developing countries were disproportionally affected by mental symptoms compared to their counterparts in developed countries [12]. The prevalence of depression and anxiety symptoms among Chinese PLWH is 61% and 43%, respectively [13]. The symptom burden may seriously affect PLWH’s ART medication adherence, clinical prognosis, and quality of life [3,8,14]. Furthermore, Chinese PLWH’s needs for symptom management are unmet due to their severe symptom burden, fear of asking for help because of HIV-related stigma, and difficulties in getting timely assistance for symptom management from medical staff [15,16]. Therefore, the development of symptom management for Chinese PLWH is urgently needed.

Smartphone use has been increasing and popularizing rapidly in the last two decades [17]. The health education and self-management strategies provided by the mobile health (mHealth) applications (apps) benefit both PLWH and medical professionals [18]. Several mHealth apps designed for PLWH provide functionalities, including medication reminders, communication to peers or providers, medical information searching engine and resources, and laboratory reports [19]. Although one app was previously developed to facilitate symptom management among PLWH [20], culturally tailored symptom management tools and suitable strategies for Chinese PLWH are still limited. Our research team developed a mHealth app to provide PLWH with a more convenient tool for personalized symptom management. This study aimed to demonstrate the development process of the symptom management (SM) app and its structure and content.

## 2. Framework

Our study was guided by a framework (Figure 1) informed by the University of California, San Francisco (UCSF) Symptom Management Model [21] and the Self-regulatory HIV/AIDS Symptom Management Model (SSMM-HIV) [14]. Three key components of symptom management are symptom experience, management strategies, and outcomes. Symptom experience focuses on symptom occurrence (the cognitive pathway) and symptom distress (the emotional pathway). Symptom management strategies consider who, what (the nature of the strategy), when, where, how (delivered), to whom (the recipient of intervention), how much (the intervention dose) and why. Outcomes involve symptom outcomes and clinical outcomes. Social support, ART medication adherence, and quality of life are equally essential components of HIV/AIDS symptom management. Guided by this framework, we designed the main functions and content of the SM app.

## 3. Materials and Methods

According to the Good Practice Guidelines on Health Apps and Smart Devices [22], it was important to collect health content information from evidence-based resources and the opinions from multidisciplinary experts (healthcare professionals, engineers, professional bodies, patient or consumer associations, etc.). Therefore, the development of the SM app included two phrases. Nursing researchers first systematically searched for evidence-based resources to summarize up-to-date evidence for symptom management and health education and then completed the app’s first draft. At the next step, the multidisciplinary research team including physicians, nurses, software engineers, and nursing researchers, evaluated the quality of the app through a group meeting.

### 3.1. Drafting the Structure and Content of the App

Our research team designed the overall structure of the SM app according to the symptom management framework and determined the symptoms based on the modified sign and symptom checklist for HIV (SSC-HIV rev) [23] and self-completed HIV symptom index [24]. These two tools reflected common symptoms of PLWH and had been applied among Chinese PLWH in previous studies [16,25]. Then nursing researchers searched health education and coping strategies for each symptom.

Our research team developed the Chinese culturally adapted AIDS Clinical Nursing Practice Guidelines in 2014, which contained comprehensive evidence for common symptoms [26]. Given this important previous work, we only searched for literature that stood in the high hierarchy of the evidence pyramid, i.e., guidelines rather than original studies and systematic reviews. We also included clinical manuals and books in both English and Chinese concerning PLWH treatment and care that focused on symptom evaluation, symptom treatment, and symptom management strategies. We thoroughly searched evidence-based resources at various websites, including the World Health Organization, UNAIDS, the Centers for Disease Control and Prevention, Association of Nurses in AIDS Care, European AIDS Clinical Society Department of Health and Human Services, U.S. Department of Health and Human Services, International Association of Providers of AIDS Care, British HIV Association, New York State Department of Health AIDS Institute, Office of the AIDS Research Advisory Council (OARAC), and Canadian AIDS Treatment Information Exchange (CATIE).

### 3.2. Group Discussion and Written Feedback from the Multidisciplinary Research Team

We drafted the structure and content of the app from the review of the evidence-based resource. A group discussion meeting with the multidisciplinary research team was conducted in November 2017 in a conference room at the Shanghai Public Health Center (SPHC) affiliated with Fudan University. The handbook and checklist that quantitatively evaluated the quality of the app were sent to each expert by email in advance. During this 150 min group discussion meeting, a researcher (the sixth author) introduced the framework and contents of the app, and experts individually gave their suggestions and feedback. A total of 119 items, including the health tracking module, the self-assessment module, coping strategies for 18 symptoms (80 items), medication management, complementary therapy, diet management, exercise, relaxation techniques, and the obtaining support module were evaluated in terms of sufficient evidence, situational suitability, practicability, cost-effectiveness, and understandability (scores from 1 to 4). The group discussion was recorded and analyzed within 24 h. Members could also communicate with the researcher staff for additional suggestions after the meeting. Informed consent was obtained from all group members.

## 4. Results

### 4.1. Suggestions and Feedback from the Group Discussion

After systematically searching and summarizing evidence-based resources for symptom management (Appendix A presents the detailed sources), we drafted the structure and content of the app and then conducted the multidisciplinary group meeting. Ten experts, including hospital managers, nursing professors, physicians, nurses, and software engineers, gave their suggestions and feedback. Among them, three had Ph.D. degrees, three had master’s degrees, and six had bachelor’s degrees. All experts completed the evaluation checklist. They generally agreed with the framework and content of the app and gave high scores on the questionnaire. Appendix A presents the detailed scores for each item. The qualitative suggestions were as follows:

The health tracking module operated convenient and attractive functions of checking laboratory tests. However, this model needed to be enhanced with data visualization. Tables or trend charts were recommended to help users track their health indicators. According to this suggestion, the revised app added the indicator trend submodule in the health tracking module. When an indicator had more than two times of data, this submodule could generate a trend chart.

Experts strongly recommended simplifying the assessment tools in the self-assessment module. For instance, we could apply the two-item Patient Health Questionnaire (PHQ-2) instead of the nine-item Patient Health Questionnaire (PHQ-9) to assess depression. According to this suggestion, all symptom assessment tools were simplified to less than four items (Appendix A).

We revised or deleted some details of the symptom management coping strategies according to the experts’ suggestions to make the strategies applicable in the local context. For example, (a) the experts thought that some complementary therapies for fatigue and diet therapies for depression were not suitable for the recommendation. Therefore, we deleted the following: “Taking vitamin or mineral supplements, such as vitamin B12, can help relieve fatigue. Some traditional Chinese medicines, such as rhodiola, ginseng, and licorice, may also help relieve fatigue.” and “We suggest consuming more vitamin-D-rich foods, such as eggs, and vitamin B, such as blueberries and bamboo shoots to ameliorate depressive symptoms. Additionally, food rich in Omega-3 and tryptophan may help improve moods.” (b) Symptom diaries (sleep diary and emotion diary) were recommended for deletion. One expert (E8) thought they were complicated and increased the amount of reading for users. (c) The deletion of some medication advice for symptom management was suggested. For example, some listed medicines for diarrhea were not available in China and should be deleted (E1 and E3). Medication for perianal neoplasms should be used under particular conditions. The listing of these medications in the app, which might be misleading and cause clinical risks, was suggested to be deleted (E10). One of the headache management strategies was as follows: “Herbal short-toned chrysanthemum may be effective in relieving headache. It is recommended to take it more than two hours apart from the ART medication and under the guidance of medical staff.” Short-toned chrysanthemum was not commonly prescribed clinically and ought to be deleted (E2). (d) One item for fever management was as follows: “Fever may occur at any stage of the disease. It is an indicator of the disease and a process by which the immune system reacts to outside pathogens. When experiencing a fever, you should focus on cooling down, identifying the cause of your fever, preventing chills, and consuming enough water.” The experts suggested revising and adding the following expression: “When experiencing a fever, you should visit a doctor as soon as possible.” (E3)

In order to provide more comprehensive medication guidance, experts recommended adding a table of ART medication interactions with other medications in the app. We accepted this suggestion and added the table in the medication management submodule in the coping strategies module.

Stigma might discourage PLWH from downloading and using this app. Logo and homepage should not show HIV. In addition, data security and confidentiality should be strengthened to reduce patients’ concerns. The following measures were conducted to address these issues: (a) We removed all HIV-related information in the app logo and homepage. (b) To ensure internal network security, we could use virtual local area network (VLAN) technology and logical isolation strategies. Using the latest firewall technology and packet filtering or proxy technologies could allow data to pass through selectively, making it possible to effectively monitor any activity between the internal network and the external network and prevent malicious and illegal access. (c) All data were recommended to be transmitted on the Web API (application programming interface) to ensure data transmission security. Encrypting sensitive data during the transmission process was recommended. An application authentication and authorization mechanism were also established. (d) Users could set a login password and/or gesture password. The screen would be locked if no operations were performed in the last 5 min. (e) To prevent data leakage, we transmitted the patient’s laboratory test data directly from the hospital’s health information system (HIS) to the mobile terminal of the app. The web-based administration portal could not acquire any user’s medical or identity information.

It would be user-friendly to provide a communication platform between medical staff and PLWH. Thus, medical staff could give personalized information support according to the patients’ questions. We accepted this suggestion and designed a question and answers (Q&As) submodule in the obtaining support module (the staff answered questions from users on the web-based administration portal). Medical staff should answer users’ questions on the web-based administration portal within 48 h.

In order to strengthen the sustainability of the SM app, experts recommended designing neat and aesthetically pleasing app interfaces and providing various forms of information, including text, pictures, audio, and videos, to attract more users. They also recommended embedding the app into the routine process of clinical follow-ups, such as case management in SPHC.

### 4.2. Structure and Key Functions of the App

Based on all our previous work, the final app structure and functions in the mobile terminal include eight modules. Users see the home page as Figure 3a after opening the SM app (Figure 2). Four major modules are shown on the home page, including health tracking, self-assessment, coping strategies, and obtaining support. Other modules with smaller logos are newly registered user, regular user, symptom history, and reminders and settings. By clicking the round blue button at the top left of the home page, users can connect their personal information with the HIS of SPHC through the Certification Information button (Figure 3c). Technicians check users’ ID numbers and names within 48 h and give user’s permission to browse their medical reports in the web-based administration portal. If users have had visits in SPHC, they can see all the information, including their prescriptions and laboratory reports, in the SM app.

#### 4.2.1. Self-Assessment Module

By clicking the self-assessment module on the home page, users can see symptom assessment options as in Figure 4a. Symptom assessment includes two mental symptoms (depression and anxiety) and 16 physical symptoms (fatigue, headache, muscle and joint pain, fever, sleep disorder, diarrhea, nausea or vomiting, shortness of breath or cough, numbness/tingling of hands or feet, perianal neoplasms, fat redistribution, weight loss, rash, oral leukoplakia or ulcers, memory loss, and blurred vision). By clicking on one of the symptom buttons (such as depression), users can see the assessment tools as in Figure 4c. We applied PHQ-2 to screen depression [27] and the Generalized Anxiety Disorder scale (GAD-2) to screen anxiety [28]. Both PHQ-2 and GAD-2 are widely used valid screening tools. There were no available validated and widely-used screening tools for each physical symptom. Therefore, we designed screening tools for 16 physical symptoms. According to the symptom management framework, symptom experience focuses on symptom occurrence (the cognitive pathway) and symptom distress (the emotional pathway), i.e., symptom severity and symptom distress. Therefore, all the physical symptoms were assessed from these two aspects. For example, the two items for fatigue: (1) If a score of 0 means no fatigue at all, and a score of 10 means the worst fatigue you can imagine, how much do you rate your fatigue over the past 2 weeks? (2) How much does fatigue affect your daily life (0 = not at all, 1 = a little, 2 = a moderate amount, 3 = very much, 4 = extremely)? All the brief self-reported tools that assess each symptom are available in Appendix A.

A total of 80 symptom management strategies are available in the SM app. Users receive personalized result interpretations and strategies (Figure 5a) according to their assessment submission and cutoffs. Appendix 3 also presents all the cutoffs for each assessment tool.

#### 4.2.2. Health Tracking Module

By clicking the health tracking module on the home page, users can see their latest prescription and four submodules on the screen, i.e., laboratory report, indicator trend, medication list, and symptoms results (Figure 5c). By clicking the laboratory report button, users can see their lab report results transferred from the HIS system (Figure 6a). If they have not visited SHPC before, they can input blood test results themselves. The indicator trend button in the health tracking module can generate tables and trend charts to help users track their health indicators, including their viral load, CD4+ T cell count, body temperature, body weight, pain, fatigue, and ART medication adherence (Figure 6c).

#### 4.2.3. Coping Strategy Module

The coping strategy module on the home page provides basic knowledge and coping strategies, including medication management (ART medication introduction, the principles of taking ART medication, taking medicine under particular circumstances, the interaction of ART medication with other medication, and coping strategies for the side effects of ART medication), the principles of using a complementary therapy, diet adjustment (balanced diet, safe diet, and diet under special circumstances), exercise (exercise principles, exercise choice, exercise plan, and precautions), and relaxation training (mindfulness, deep breathing, image guidance, body scan, and activating events–beliefs–consequences (ABC) rational-emotive behavior therapy).

#### 4.2.4. Obtaining Support Module

The obtaining support module on the home page has four submodules including, health care support, peer support, information support, and Q&As. Any questions left on the Q&As submodule are answered by medical staff within 48 h.

#### 4.2.5. Other Modules

The newly registered user module on the home page provides knowledge patients who took ART medication for less than six months needed to know, including an introduction to HIV, an introduction to ART, ART medication adherence, timely follow-up, safety measures, and a healthy lifestyle. The regular user module on the home page provides knowledge and topics designed for patients who have taken ART medication for more than six months, including treatment goals, treatment progress, medication resistance, medication side effects, healthy lifestyle, and fertility options.

#### 4.2.6. Web-Based Administration Portal Functions

The web-based administration portal for medical staff and technical staff includes five functions:The user management function manages all registered account information.The information publishing function can push or edit health education content.The content management function can summarize user assessment data and provide reminders for outliers. Case managers contact and provide personalized interventions based on their lab tests.The Q&A function can answer user questions.Other functions include basic statistical functions such as analyzing users’ login numbers and viewing each module’s number.

### 4.3. Privacy and Confidentiality

The following design helps users understand that the SM app protects users’ privacy and confidentiality. Firstly, people cannot distinguish from the logo (Figure 2) and homepage (Figure 3a) that the SM app was designed for PLWH. Secondly, it is not mandatory for users to provide their personal information. If they do not provide their ID number, they can still use all the functions of the app. The only advantage for connecting their ID number with our hospital is that they can check and look up all the medical prescriptions, laboratory tests through the app as Figure 5c and Figure 6a. If users do not provide personal information but still want to see the generated trend of their health status based on the health indicators, such as CD4 + T cell count (Figure 6c), they can input data by themselves. Thirdly, users can set a login password and/or guest password. The screen is locked if no operations are performed in the last 5 min.

## 5. Discussion

To our knowledge, the SM app was the first attempt to develop a mHealth app for personalized symptom management for PLWH in China. It was developed based on evidence-based resource reviews and input from our multidisciplinary research team. The SM app can be operated on both Android and iOS systems. It also has a web-based administration portal managed by medical staff and technicians.

mHealth technology has been developing rapidly in recent years. Given the potential to improve access to care by reducing geographical and financial barriers, mHealth apps provide feasible, accessible, and effective platforms for self-management of many chronic diseases, including hypertension [29], diabetes [30], and cancer [31]. They are also increasingly being used for the care of PLWH [19]. HIV infection is a lifelong chronic disease. After HIV infection, symptom management is a long-term task for PLWH. PLWH also have different symptom prevalence and severity. Therefore, mHealth apps are promising choices for expanding the HIV personalized symptom management interventions.

The SM app provides reliable symptom management knowledge from evidence-based resources that are recommended by researchers. The multidisciplinary group discussion was an efficient way to collect opinions from different areas and perspectives and played an important role in ensuring the accuracy and cultural adaptability of the app’s content and promoting the app’s acceptability. After the input from the multidisciplinary research group, the app could meet PLWH’s personalized needs for symptom management and became a promising tool to promote case management for medical professionals.

One symptom management app for PLWH in the United States of America (mVIP) provides symptom management strategies for 13 symptoms [20]. Our app has a bigger symptom pool (18 symptoms), which may cover more PLWH’s symptom issues. Guided by a symptom management framework, the SM app also includes health contents and functions about improving medication adherence and social support. General health contents are provided for newly registered users and regular users and may improve the quality of life outcome. Particularly, the SM app has several personalized symptom management functions. Firstly, PLWH can assess their symptoms and receive different symptom management strategies according to their assessment results. Secondly, the health tracking module provides PLWH with a convenient tool to track their health indicators. Thirdly, PLWH can communicate with medical staff to gain support through the SM app. These personalized functions can meet personalized needs and may encourage more usage.

Several limitations of this study should be noted. Firstly, the SM app currently only connects with the HIS of the SPHC, and users can only check prescriptions and laboratory tests from SPHC. Secondly, PLWH did not participate in the development process of the app. Future research is needed to examine whether the SM app meets the expectations of PLWH and medical staff and evaluate the app’s usability. The app will be revised and updated according to the user feedback. When the SM app is determined to be user-friendly and suitable to be promoted to a larger group of PLWH, its effects on symptom management for PLWH will be tested and open access to HIS in other hospitals.

## 6. Conclusions

The SM app was developed based on evidence-based resource reviews and the input from our multidisciplinary research team. It provided PLWH with reliable symptom management knowledge and personalized symptom management functions. The web-based administration portal also provided the medical staff with convenient functions for case management. Future studies are needed to further test the app’s usability and effectiveness on symptom management. We will refine the SM app based on users’ experience and the collected data.

## 7. Patents

The SM app, which was developed for both iOS and Android, applied for a patent (National Copyright Administration of the People’s Republic of China: No.02700189).

## Figures and Tables

**Figure 1 jpm-11-00346-f001:**
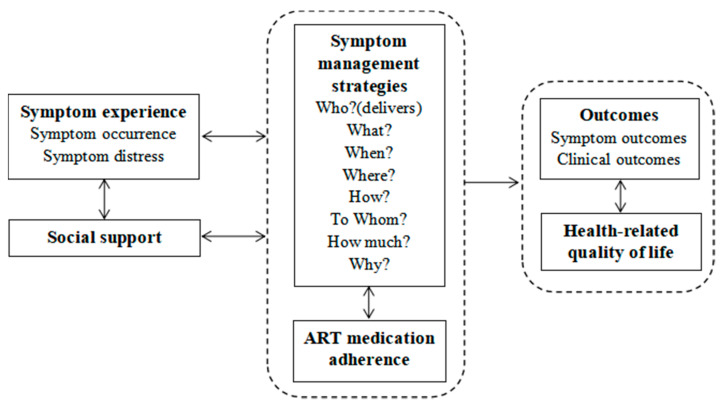
Symptom Management Framework.

**Figure 2 jpm-11-00346-f002:**
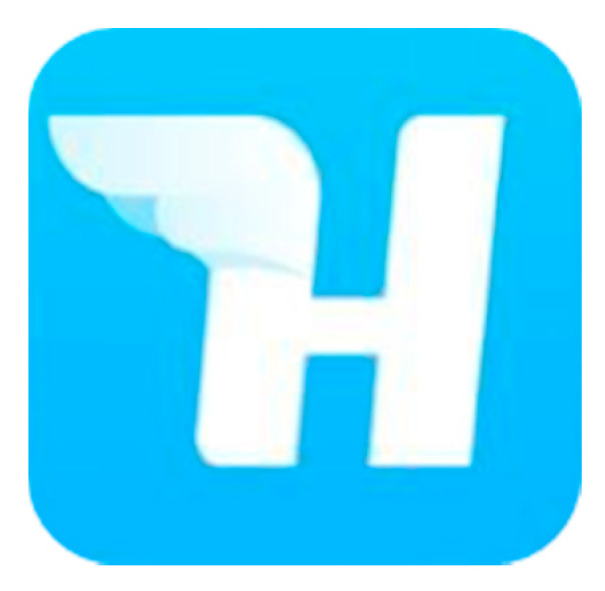
Logo of the SM app.

**Figure 3 jpm-11-00346-f003:**
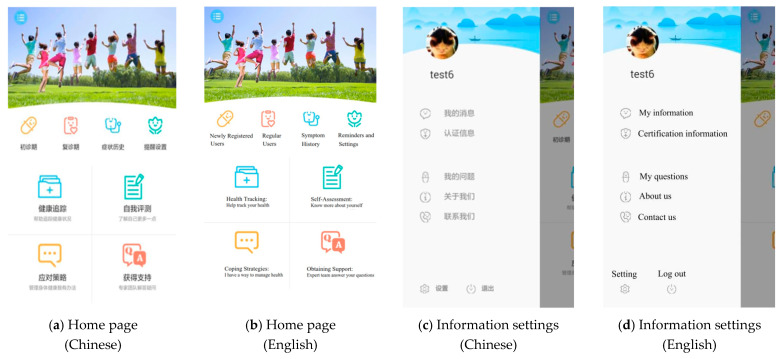
Screenshot of the home page and information settings.

**Figure 4 jpm-11-00346-f004:**
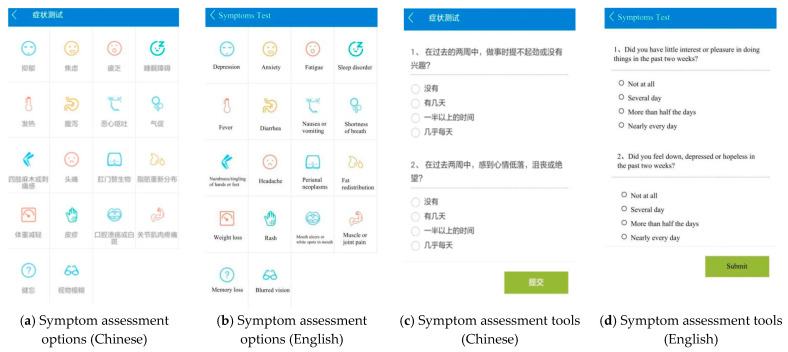
Screenshot of symptom assessment options and symptom assessment tools.

**Figure 5 jpm-11-00346-f005:**
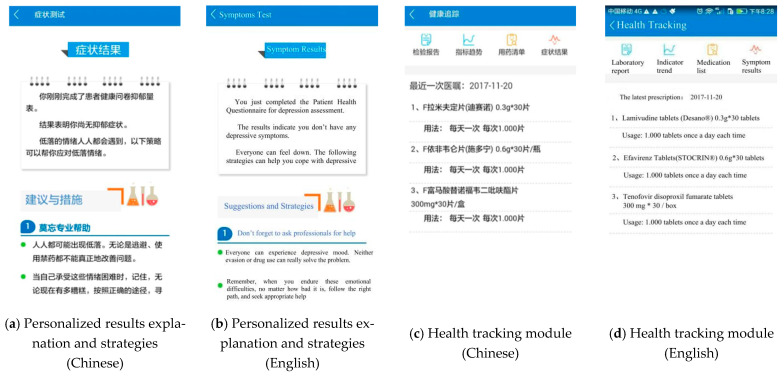
Personalized results explanation and strategies, and the health tracking module.

**Figure 6 jpm-11-00346-f006:**
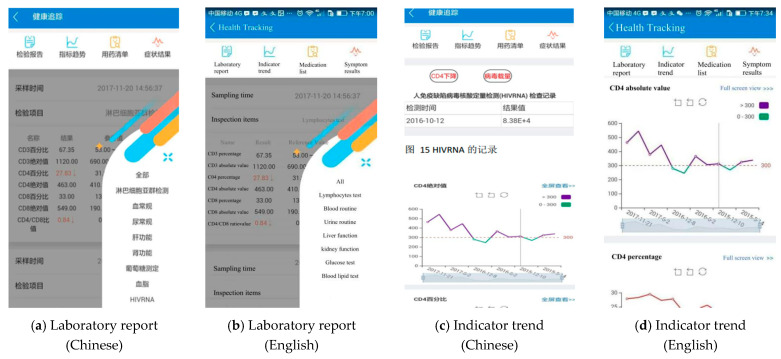
Screenshot of laboratory report and indicator trend.

## Data Availability

All the raw data are available in the Appendix A.

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
