# Peer review of "The Development of a Personalized Symptom Management Mobile Health Application for Persons Living with HIV in China"

_jpm, 2021, doi:10.3390/jpm11050346_

Round 1

Reviewer 1 Report

This article on SM app development is an important tool for PLIH target population and would allow improve quality of care through the technological innovations through mobile health interventions. However, the developmental phase of the SM app requires substantive improvements and especially a thorough revision on the methods section among other areas. Please find the comments below:

Overall comment: The English of the paper needs to be revised and checked for grammatical errors.

Line 56-57: It would be important to know more about the National ART Strategy (e.g. the Four Frees and one care for HIV/AIDS) that would allow readers to know why this mobile application is important for this population group (PLIH).

Line 65-66: Why the symptom management needs for PLIH are unmet in China? Please provide what the contextual factors, situational factors and behavioural factors involved ? This would allow the reader to steer attention to the development of mobile application resource needs.

Line 88-95: It would be important to understand what were the contents of health education that were searched? It would be important to provide examples how the information were compared as not all symptom management would apply to a Chinese PLIH population context. It is also important for the authors to describe if any systemic literature search were performed to collect information from the literature. What were the strategies put in place to include/exclude health information for the development of the app? The authors should mention what did they find from their searches and how it aligns/tailored to the app development process. It is unclear how and why authors chose the health contents for the SM app.

Figure (b): It is unclear what the age group is for junior or senior users. Please clarify.

Figure 3: The options provided in the app, it should be explained whether these items are the standard symptom management items that are Applicable for PLIH in China. 

Symptom Test in Figure 4 (b): There is a need to understand the reliability and specificity of all these question items. E.g. is question item e.g. PHQ-2 is reliable? How only two question items can identify/confirm a person as "depressed"? Authors should clarify the process further. The authors should also provide details on how they carried out the question item checklist to be included in the SM app? This should be included in the methods section and revise the methods section.

Figure 5 (b): The statement "you do not have any depressive symptoms" needs to be rephrased. A user may not completely find this statement as a satisfactory response as users may feel underlying emotional distress that the app may not capture. In addition, the authors needs to explain how many users they have tested these 75 symptom strategies items if at all. If not, then provide a detailed process/plan on how to carry them out.

In terms of user confidentiality, it is important the authors mention how the users will understand about their privacy and confidentiality? What steps would be followed and how users can provide/receive information about user confidentiality. 

Author Response

This article on SM app development is an important tool for PLIH target population and would allow improve quality of care through the technological innovations through mobile health interventions. However, the developmental phase of the SM app requires substantive improvements and especially a thorough revision on the methods section among other areas. Please find the comments below:

Overall comment: The English of the paper needs to be revised and checked for grammatical errors.

Responses: We appreciate these reviewers’ valuable comments. These comments will help strengthen our manuscript. We carefully addressed each point raised by the reviewers, and revised the text accordingly. We also asked a native English speaker to check for grammatical errors. Below please find our detailed responses to each of the comments.

Line 56-57: It would be important to know more about the National ART Strategy (e.g. the Four Frees and one care for HIV/AIDS) that would allow readers to know why this mobile application is important for this population group (PLIH).

Responses: We thank the reviewer for the comment. We have added the following sentences in the introduction.

“Chinese PLWH receive free ART based on the National Strategies of “Four Frees and One Care for HIV/AIDS” [9]. However, only publicly-funded domestic antiretroviral medications are free of charge. The integrase strand transfer inhibitors (INSTIs), which are the internationally recommended first-line treatment medication for PLWH [10], by contrast, are imported and needed to be paid out-of-pocket. Therefore, due to the issue of affordability, instead of taking these imported INSTIs, most Chinese PLWH choose to take free ART medications, e.g., efavirenz (EFV) or lopinavir/ritonavir (LPV/r). These medications may increase their risks for medication toxicity and symptoms distress (EFV may lead to rash and sleep disorders, LPV/r may lead to diarrhea [11]).” (Page 2 Line 56-65)

Line 65-66: Why the symptom management needs for PLIH are unmet in China? Please provide what the contextual factors, situational factors and behavioural factors involved ? This would allow the reader to steer attention to the development of mobile application resource needs.

Responses: We thank the reviewer for the comment. We have added the following information in the introduction.

“Furthermore, Chinese PLWH’s needs for symptom management are highly unmet due to their severe symptom burden, fear of asking for help because of HIV-related stigma, and difficulties in getting timely assistance for symptom management from medical staff [15,16]. Therefore, the development of symptom management interventions for Chinese PLWH is in urgent need..” (Page 2 Line 70-74)

Line 88-95: It would be important to understand what were the contents of health education that were searched? It would be important to provide examples how the information were compared as not all symptom management would apply to a Chinese PLIH population context. It is also important for the authors to describe if any systemic literature search were performed to collect information from the literature. What were the strategies put in place to include/exclude health information for the development of the app? The authors should mention what did they find from their searches and how it aligns/tailored to the app development process. It is unclear how and why authors chose the health contents for the SM app.

Responses: We thank the reviewer for the comments. All the 18 symptoms involved in the app came from a modified sign and symptom check-list for HIV (SSC-HIV rev) [1] and self-completed HIV symptom index [2]. We searched health education and coping strategies for each symptom. Since our research team have developed the Chinese culturally adapted AIDS Clinical Nursing Practice Guidelines in 2014 that contained comprehensive evidence of common symptoms [3], we only searched guidelines rather than original studies and systematic reviews. After extracting health education content of each symptom from evidence-based resources, all members in our multidisciplinary research team provided input to ensure these symptoms management strategies were applicable in local context. Our previous work (the development of Chinese culturally adapted AIDS Clinical Nursing Practice Guidelines) provided a scientific foundation for the cultural adaption process of this intervention. We made further revisions based on the input from our multidisciplinary research team. For example, we deleted several medications for diarrhea in the medication management strategies because they are not listed and are not available in China.

We have clarified these issues in the revised manuscript.

“Our research team designed the overall structure of the app according to a symptom management framework and determined the symptoms based on the modified sign and symptom check-list for HIV (SSC-HIV rev) [23] and self-completed HIV symptom index [24]. These two tools reflect common symptoms of PLWH, and have been applied among Chinese PLWH in previous studies [16,25]. Then nursing researchers searched health education and coping strategies for each symptom.” (Page 3 Line 111-116)

“Our research team has developed the Chinese culturally adapted AIDS Clinical Nursing Practice Guidelines in 2014, which contained comprehensive evidence for common symptoms [26]. Given this important previous work, we only searched for literature that stands in high hierarchy of the evidence pyramid, i.e., guidelines rather than original studies and systematic reviews. We also included clinical manuals and books in both English and Chinese concerning PLWH treatment and care that focus on symptom evaluation, symptom treatment, and symptom management strategies.” (Page 3 Line 117-123)

“A total of 119 items, including the health tracking module, the self-assessment module, coping strategies for 18 symptoms (80 items), medication management, complementary therapy, diet management, exercise, relaxation techniques, and the obtaining support module, were evaluated in terms of sufficient evidence, situational suitability, practicability, cost-effectiveness, and understandability (scores from 1 to 4).” (Page 4 Line 139-144)

“We revised or deleted some details of the symptom management coping strategies according to the experts’ suggestions to make the strategies more applicable in the local context. For example......” (Page 4 Line 169-171)

Figure (b): It is unclear what the age group is for junior or senior users. Please clarify.

Responses: We apologize for this error. We have corrected the spelling for this figure.

Figure 3: The options provided in the app, it should be explained whether these items are the standard symptom management items that are Applicable for PLIH in China.

Responses: We thank the reviewer for the comment. As we explained earlier, all the 18 symptoms involved in our app came from the modified sign and symptom check-list for HIV (SSC-HIV rev) [1] and self-completed HIV symptom index [2]. These two tools are widely used, reflecting common symptoms of PLWH, and have been applied to Chinese PLWH [4,5]. Our university affiliated hospital has also implemented this checklist as one of the tools for case management. Therefore, we believe that these symptom items are applicable for PLWH in China. We also have clarified this issue in the revised manuscript.

“Our research team designed the overall structure of the app according to the symptom management framework and determined the symptoms based on the modified sign and symptom check-list for HIV (SSC-HIV rev) [24] and self-completed HIV symptom index [25]. These two tools reflect common symptoms of PLWH, and have been applied among Chinese PLWH in previous studies [16,26].” (Page 3 Line 111-115)

Symptom Test in Figure 4 (b): There is a need to understand the reliability and specificity of all these question items. E.g. is question item e.g. PHQ-2 is reliable? How only two question items can identify/confirm a person as "depressed"? Authors should clarify the process further. The authors should also provide details on how they carried out the question item checklist to be included in the SM app? This should be included in the methods section and revise the methods section.

Responses: We thank the reviewer for the comments. Firstly, considering the readability and usability of the app, we chose and designed brief tools for all 18 symptoms. Secondly, these tools were used to screen for symptoms, rather than for diagnosis. Thirdly, PHQ-2 only has 2 items. However, it is a widely used tool for depression screening [6]. So does the screening tool for anxiety (GAD-2) [7]. There were no available validated and widely-used screening tools for each physical symptom. Therefore, we designed screening tools for 16 physical symptoms based on the suggestions from the health professionals. The design was based on the University of California, San Francisco (UCSF) Symptom Management Model [8]. According to this model, symptom experience focuses on symptom occurrence (the cognitive pathway) and symptom distress (the emotional pathway, i.e., symptom severity and symptom distress). Therefore, all the physical symptoms were assessed from these two aspects. For example, the two items for fatigue are: (1) If a score of 0 means no fatigue at all, and a score of 10 means the worst fatigue you can imagine, how much do you rate your fatigue over the past 2 weeks? (2) How much does fatigue affect your daily life? (0 = not at all, 1 = a little, 2 = a moderate amount, 3 = very much, 4 = extremely).

   We also have clarified this issue in the revised manuscript.

“We applied PHQ-2 to screen depression [27] and the Generalized Anxiety Disorder scale (GAD-2) to screen anxiety [28]. Both PHQ-2 and GAD-2 are widely used valid screening tools. There were no available validated and widely-used screening tools for each physical symptom. Therefore, we designed screening tools for 16 physical symptoms. According to the symptom management framework, symptom experience focuses on symptom occurrence (the cognitive pathway) and symptom distress (the emotional pathway), i.e., symptom severity and symptom distress. Therefore, all the physical symptoms were assessed from these two aspects. For example, the two items for fatigue are: (1) If a score of 0 means no fatigue at all, and a score of 10 means the worst fatigue you can imagine, how much do you rate your fatigue over the past 2 weeks? (2) How much does fatigue affect your daily life? (0 = not at all, 1 = a little, 2 = a moderate amount, 3 = very much, 4 = extremely).” (Page 6 Line 251-263)

Figure 5 (b): The statement "you do not have any depressive symptoms" needs to be rephrased. A user may not completely find this statement as a satisfactory response as users may feel underlying emotional distress that the app may not capture. In addition, the authors needs to explain how many users they have tested these 80 symptom strategies items if at all. If not, then provide a detailed process/plan on how to carry them out.

Responses: We thank the reviewer for the comments. As we explained above, these brief tools were used to screen for symptoms. If a user’s PHQ-2 score is 0, i.e., in the past 2 weeks, he/she reported never feeling the following symptoms: little interest or pleasure in doing things, feeling down, and depressed or hopeless, we would consider he/she might not have depressive symptoms. The SM app serve as a tool for symptoms self-management. Therefore, we used this statement.

  As for the second question, the app has 80 symptom strategies for 18 symptoms. We do not think users need to browse all of them at one time. As we describe in the introduction, the average number of symptoms among PLWH is 8-17 [9–11]. If PLWH do not have some symptoms, they may not assess these symptoms and browse relevant strategies. This app provides symptom screening tools and resources. Users can assess symptoms and browse strategies according to their needs and interests. Although we don’t have the expectation that every user will browse every strategy, the app provides a clear and vivid interface for all the symptom assessment options, which we believe can attract user’s interest and encourage them assess more symptoms and read more strategies.

In terms of user confidentiality, it is important the authors mention how the users will understand about their privacy and confidentiality? What steps would be followed and how users can provide/receive information about user confidentiality. 

Responses: We thank the reviewer for the comments. The following design will help users understand the SM app will protect users’ privacy and confidentiality. Firstly, people cannot distinguish from the logo and homepage that the SM app is designed for PLWH.

Secondly, it is not mandatory for users to provide their personal information. If they do not provide their ID number, they can still use all the functions of the app. The only advantage for connecting their ID number with our hospital is that they can check and look up all the medical prescriptions, laboratory tests through the app as illustrated below.

If they do not provide personal information but still want to see the generate trend of their health status based on the health indicators, such as CD4+T cell count, as follows, they can input data by themselves.

Thirdly, users can set a login password and/or guest password. The screen will be locked if no operations are performed in the last 5 minutes.

The second question: as we described in the manuscript, users just need to click the Certification Information button, and fill out their name and ID number. Technicians will check their ID number and name within 48 hours, and give users’ permission to browse their medical reports in the web-based administration portal. If users have had medical visits in the hospital, they can see all the information, including their prescriptions and laboratory reports in the SM app.

“Click the round blue button at the top left of the home page, and users can connect their personal information with the HIS system of SPHC through the Certification Information button (Figure 2 [c]). Technicians check users’ ID numbers and names within 48 hours and give users’ permission to browse their medical reports in the web-based administration portal. If users have had medical visits in SPHC, they can see all the information, including their prescriptions and laboratory reports, in the SM app.” (Page 6 Line 235-240)

References

  1. 1. Holzemer, W.L.; Hudson, A.; Kirksey, K.M.; Hamilton, M.J.; Bakken, S. The Revised Sign and Symptom Check-List for HIV (SSC-HIVrev). J Assoc Nurses AIDS Care2001, 12, 60–70, doi:10.1016/s1055-3290(06)60263-x.
  2. 2. Justice, A.C.; Holmes, W.; Gifford, A.L.; Rabeneck, L.; Zackin, R.; Sinclair, G.; Weissman, S.; Neidig, J.; Marcus, C.; Chesney, M.; et al. Development and Validation of a Self-Completed HIV Symptom Index. J Clin Epidemiol2001, 54 Suppl 1, S77-90, doi:10.1016/s0895-4356(01)00449-8.
  3. 3. Fu, L. Development of the HIV/AIDS Clinical Nursing Practice Guidelines, Fudan University: Shanghi, 2014.
  4. 4. Zhu, Z.; Hu, Y.; Guo, M.; Williams, A.B. Urban and Rural Differences: Unmet Needs for Symptom Management in People Living With HIV in China. J Assoc Nurses AIDS Care2019, 30, 206–217, doi:10.1097/JNC.0000000000000025.
  5. 5. Zhu, Z.; Hu, Y.; Xing, W.; Guo, M.; Zhao, R.; Han, S.; Wu, B. Identifying Symptom Clusters Among People Living With HIV on Antiretroviral Therapy in China: A Network Analysis. Pain Symptom Manage.2019, 57, 617–626, doi:10.1016/j.jpainsymman.2018.11.011.
  6. 6. Kroenke, K.; Spitzer, R.L.; Williams, J.B.W. The Patient Health Questionnaire-2: Validity of a Two-Item Depression Screener. Med Care2003, 41, 1284–1292, doi:10.1097/01.MLR.0000093487.78664.3C.
  7. 7. Kroenke, K.; Spitzer, R.L.; Williams, J.B.W.; Monahan, P.O.; Löwe, B. Anxiety Disorders in Primary Care: Prevalence, Impairment, Comorbidity, and Detection. Intern. Med.2007, 146, 317–325, doi:10.7326/0003-4819-146-5-200703060-00004.
  8. 8. Dodd, M.; Janson, S.; Facione, N.; Faucett, J.; Froelicher, E.S.; Humphreys, J.; Lee, K.; Miaskowski, C.; Puntillo, K.; Rankin, S.; et al. Advancing the Science of Symptom Management. J Adv Nurs2001, 33, 668–676, doi:10.1046/j.1365-2648.2001.01697.x.
  9. 9. Chen, W.-T.; Shiu, C.; Yang, J.P.; Tun, M.M.M.; Zhang, L.; Wang, K.; Chen, L.-C.; Aung, M.N.; Lu, H.; Zhao, H. Tobacco Use and HIV Symptom Severity in Chinese People Living with HIV. AIDS Care2020, 32, 217–222, doi:10.1080/09540121.2019.1620169.
  10. 10. Lee, K.A.; Gay, C.; Portillo, C.J.; Coggins, T.; Davis, H.; Pullinger, C.R.; Aouizerat, B.E. Symptom Experience in HIV-Infected Adults: A Function of Demographic and Clinical Characteristics. J Pain Symptom Manage2009, 38, 882–893, doi:10.1016/j.jpainsymman.2009.05.013.
  11. 11. Olson, B.; Vincent, W.; Meyer, J.P.; Kershaw, T.; Sikkema, K.J.; Heckman, T.G.; Hansen, N.B. Depressive Symptoms, Physical Symptoms, and Health-Related Quality of Life among Older Adults with HIV. Qual Life Res2019, 28, 3313–3322, doi:10.1007/s11136-019-02271-0.

Reviewer 2 Report

Thank you for the opportunity to review this article describing the development of an app to manage HIV symptoms of consumers in China.  While the paper is easy to read an follows logical progression and figures are well described there are a number of suggestions to strengthen the article.

In the abstract (L28+) the information in brackets is important and they need to be removed.

In the main body of the article I suggest the authors go through and for consistency ensure they have put all acronyms (WEBAPI) etc and abbreviations (Q&A etc) in full and/or provide an explanation for those they do not give any information on – readers who are not familiar with all the HIV medication regiments or Chinese encryption protocols may not be aware of the terms used or indeed the herbal remedies mentioned ie short-tones? An international readership may use other terms, so more detail may be required to enable full understanding of the app and/or principles involved (to enable replicability) and system management.

P2L69 needs to be reworded, smartphones are not growing – smartphone use is growing, similarly health information is provided by information on the app software.  These sentences need to be reworded to clearly reflect what is meant.

P2L79 onwards.  To give full merit to the work undertaken and to enable replication for other illness or diseases, there needs to be more detail provide in the method and materials section. The opening paragraph is vague.  There is no reference to any literature, although I am sure this method has been documented elsewhere. The process needs to be framed with theoretical underpinnings as there has been much app development and also though this is about HIV, there have been many apps developed for a range of disorders.  The process of this development needs to be clearly explained and demonstrate why it was so useful for this disease symptom management process.

Sub-heading may also sign post the reader regarding the decision-making of the app to enable navigation.  Additionally, there may need to be a section on the ‘human’ elements and also the software design.  There are published frameworks the authors could use to improve the robustness of their process.

For all the work undertaken the discussion is short and the conclusion is scant.  I suggest the authors address the findings and justify the results in relation to the literature in the discussion.  Currently the conclusion is one sentence.  A conclusion generally provides a summary, including the main findings.  Additionally, limitations and future directions sections could also provide the reader with the authors beliefs about the app and where to next with it.

Author Response

Thank you for the opportunity to review this article describing the development of an app to manage HIV symptoms of consumers in China. While the paper is easy to read an follows logical progression and figures are well described there are a number of suggestions to strengthen the article.

Responses: We appreciate the reviewer ‘s encouragement and these comments. These comments are valuable and will help us strengthen our manuscript. We carefully addressed each point raised by the reviewer and revised the text accordingly. Below please find our detailed responses to each of the comments.

In the abstract (L28+) the information in brackets is important and they need to be removed.

Responses: We thank the reviewer for the comment. We have deleted the sentence in brackets.

In the main body of the article I suggest the authors go through and for consistency ensure they have put all acronyms (WEBAPI) etc and abbreviations (Q&A etc) in full and/or provide an explanation for those they do not give any information on – readers who are not familiar with all the HIV medication regiments or Chinese encryption protocols may not be aware of the terms used or indeed the herbal remedies mentioned ie short-tones? An international readership may use other terms, so more detail may be required to enable full understanding of the app and/or principles involved (to enable replicability) and system management.

Responses: We thank the reviewer for this comment. We have put acronyms and abbreviations in full in the manuscript as follows.

“All data were recommended to be transmitted on the Web API (application programming interface) to ensure the data transmission security.” (Page 5 Line 208-210)

“We accepted this suggestion and designed a questions and answers (Q&As) submodule in the obtaining support module (the staff answer questions from users on the web-based administration portal).” (Page 5 Line 220-222)

P2L69 needs to be reworded, smartphones are not growing – smartphone use is growing, similarly health information is provided by information on the app software.  These sentences need to be reworded to clearly reflect what is meant.

Responses: We thank the reviewer for the correction. We have revised these sentences as follows:

“Smartphone use is increasing and popularizing rapidly in the last two decades [17]. The health education and self-management strategies provided by the mobile health (mHealth) applications (apps) benefit both PLWH and medical professionals [18]. ” (Page 2 Line 75-77)

P2L79 onwards. To give full merit to the work undertaken and to enable replication for other illness or diseases, there needs to be more detail provide in the method and materials section. The opening paragraph is vague. There is no reference to any literature, although I am sure this method has been documented elsewhere. The process needs to be framed with theoretical underpinnings as there has been much app development and also though this is about HIV, there have been many apps developed for a range of disorders. The process of this development needs to be clearly explained and demonstrate why it was so useful for this disease symptom management process.

Responses: We thank the reviewer for the comments. We have revised the methods as follows.

“According to the Good Practice Guidelines on Health Apps and Smart Devices [22], it is important to collect health content information from evidence-based resources, and the opinions from multidisciplinary experts (healthcare professionals, engineers, professional bodies, patient or consumer associations, etc.). Therefore, the development of the app included two phrases. Nursing researchers first systematically searched for evidence-based resources to summarize up-to-date evidence for symptom management and health education, and then completed the app’s first draft. At the next step, the multidisciplinary research team, included physicians, nurses, software engineers, and nursing researchers, evaluated the quality of the app through a group meeting.” (Page 3 Line 101-109)

Sub-heading may also sign post the reader regarding the decision-making of the app to enable navigation. Additionally, there may need to be a section on the ‘human’ elements and also the software design. There are published frameworks the authors could use to improve the robustness of their process.

Responses: We thank the reviewer for the comments. We have added sub-headings for modules of the SM app. We also have clarified the theoretical basis of the app framework.

“Our study is guided by a framework (Figure 1) informed by the University of California, San Francisco (UCSF) Symptom Management Model [21] and the Self-regulatory HIV/AIDS Symptom Management Model (SSMM-HIV) [14]. Three key components of symptom management are symptom experience, management strategies, and outcomes. Symptom experience focuses on symptom occurrence (the cognitive pathway) and symptom distress (the emotional pathway). Symptom management strategies consider who, what (the nature of the strategy), when, where, how (delivered), to whom (the recipient of intervention), how much (the intervention dose) and why. Outcomes involve symptom outcomes and clinical outcomes. Social support, ART medication adherence, and quality of life are equally essential components of HIV/AIDS symptom management. Guided by this framework, we designed the main functions and content of the mHealth app.” (Page 2-3 Line 87-99)

Figure 1. Symptom Management Framework

For all the work undertaken the discussion is short and the conclusion is scant.  I suggest the authors address the findings and justify the results in relation to the literature in the discussion.  Currently the conclusion is one sentence.  A conclusion generally provides a summary, including the main findings.  Additionally, limitations and future directions sections could also provide the reader with the authors beliefs about the app and where to next with it.

Responses: We thank the reviewer for the comment. We have expanded the discussion and the conclusion as follows.

“5.Discussion

   To our knowledge, the SM app is the first attempt to develop a mHealth app of personalized symptom management for PLWH in China. It was developed based on evidence-based resource reviews and input from our multidisciplinary research team. The SM app can be operated on both Android and iOS systems. It also has a web-based administration portal managed by medical staff and technicians.

mHealth technology is developing rapidly in recent years. Given the potential to improve access to care by reducing geographical and financial barriers, mHealth apps have provided feasible, accessible, and effective platforms for self-management of many chronic diseases, including hypertension [29], diabetes [30], and cancer [31]. They are also increasingly being used for the care of PLWH [19]. HIV infection is a lifelong chronic disease. After HIV infection, symptom management is a long-term task for PLWH. PLWH also have different symptom prevalence and severity. Therefore, mHealth apps are promising for expanding the HIV personalized symptom management interventions.

The SM app provides reliable symptom management knowledge from evidence-based resources that are recommended by researchers. The multidisciplinary group discussion is an efficient way to collect the opinions from different areas and perspectives and plays an important role in ensuring the accuracy and cultural adaptability of app’s content and promoting app’s acceptability. After the input from the multidisciplinary research group, the app can meet PLWH’s personalized needs for symptom management, and become a promising tool to promote case management for medical professionals.

One symptom management app for PLWH in the U.S. (mVIP) provides symptom management strategies for 13 symptoms [20]. Our app has a bigger symptom pool (18 symptoms), which may cover more PLWH’s symptom issues. Guided by a symptom management framework, the SM app also includes health contents and functions about improving medication adherence and social support. General health contents are provided for newly registered users and regular users, and may improve the quality of life outcome. Particularly, the SM app has several personalized symptom management functions. Firstly, PLWH can assess their symptoms and receive different symptom management strategies according to their assessment results. Secondly, the health tracking module provides PLWH with a convenient tool to track their health indicators. Thirdly, PLWH can communicate with medical staff to gain support through the SM app. These personalized functions can meet personalized needs and may encourage more usage.

Several limitations of this study should be noted. Firstly, the SM app currently only connects with the HIS of the SPHC, and users can only check prescriptions and laboratory tests from SPHC. Secondly, PLWH did not participate in the development process of the app. Future research is needed to examine whether the SM app meets expectations of PLWH and medical staff, and evaluate the app’s usability. The app will be revised and updated according to users’ feedback. When the SM app is determined to be user-friendly and suitable to be promoted to a larger group of PLWH, its effects on symptom management for PLWH will be tested and open access to HIS in other hospitals.

  1. Conclusions

The SM app is a promising and flexible tool for HIV symptom management. It not only provides PLWH with personalized symptom management strategies, but also provides medical staff with convenience in case management. Future studies are needed to further test the app’s usability and its effects on symptom management.” (Page 9-10 Line 315-364)

Round 2

Reviewer 1 Report

Please provide a confidentiality statement for the users in the manuscript as explained by the authors comments.

Author Response

Please provide a confidentiality statement for the users in the manuscript as explained by the authors comments.

Responses: We thank the reviewer for the comment. We have added the following confidentiality statement in the revised manuscript.

“4.3 Privacy and confidentiality

The following design will help users understand that the SM app will protect users’ privacy and confidentiality. Firstly, people cannot distinguish from the logo (Figure 2) and homepage (Figure 3 [a]) that the SM app is designed for PLWH. Secondly, it is not mandatory for users to provide their personal information. If they do not provide their ID number, they can still use all the functions of the app. The only advantage for connecting their ID number with our hospital is that they can check and look up all the medical prescriptions, laboratory tests through the app as Figure 5 (c) and Figure 6 (a). If users do not provide personal information but still want to see the generate trend of their health status based on the health indicators, such as CD4+T cell count (Figure 6 [c]), they can input data by themselves. Thirdly, users can set a login password and/or guest password. The screen will be locked if no operations are performed in the last 5 minutes.” (Page 9 Line 325-336)

Reviewer 2 Report

Thank you for the opportunity to review this revised manuscript. Thank you for the detailed letter addressing the suggestions. The attention to the revision of tenses to improve readability has accomplished the task.  There are a further couple of suggestions to strengthen the article.

In the abstract, the information in the brackets you have deleted – can be included in the abstract, just do not use the brackets -as the information is important and the brackets are not required.

P2L75, remove ‘highly’, unmet is sufficient.

P9L351, if you mean the United States of America, it needs to be in full (if first time) or the USA.

P10 line 376 onwards – conclusions.  This section still needs to be expanded and reduce the double negatives in the added sentence.  The conclusion is your opportunity to summarise the project and state the main findings.

Author Response

Thank you for the opportunity to review this revised manuscript. Thank you for the detailed letter addressing the suggestions. The attention to the revision of tenses to improve readability has accomplished the task. There are a further couple of suggestions to strengthen the article.

Responses: We appreciate the reviewer’s encouragement and these comments. Your previous comments helped us strengthen the manuscript. Again, we carefully addressed each point and revised the text accordingly. Below please find our detailed responses to each of the comments.

In the abstract, the information in the brackets you have deleted – can be included in the abstract, just do not use the brackets -as the information is important and the brackets are not required. 

Responses: We thank the reviewer for the comment and apologize for our prior misunderstanding. We have revised the sentence as follows:

“Both quantitative data and qualitative results were collected at a group discussion meeting. Quantitative data were scores of sufficient evidence, situational suitability, practicability, cost-effectiveness, and understandability (ranged from 1 to 4) for 119 items of the app contents, including the health tracking module, the self-assessment module, coping strategies for 18 symptoms (80 items), medication management, complementary therapy, diet management, exercise, relaxation techniques, and the obtaining support module.” (Page 1 Line 29-34)

P2L75, remove ‘highly’, unmet is sufficient.

Responses: We thank the reviewer for correcting this grammatical error. We have removed “highly” in this sentence.

“Furthermore, Chinese PLWH’s needs for symptom management are unmet due to their severe symptom burden, fear of asking for help because of HIV-related stigma, and difficulties in getting timely assistance for symptom management from medical staff [15,16]. ” (Page 2 Line 75-79)

P9L351, if you mean the United States of America, it needs to be in full (if first time) or the USA.

Responses: We thank the reviewer for the comment. We have corrected this spelling in the revised manuscript.

“One symptom management app for PLWH in the United States of America (mVIP) provides symptom management strategies for 13 symptoms [20]. ” (Page 10 Line 360-361)

P10 line 376 onwards – conclusions. This section still needs to be expanded and reduce the double negatives in the added sentence. The conclusion is your opportunity to summarize the project and state the main findings.

Responses: We thank the reviewer for the comment. We have revised the conclusion as follows.

“The SM app was developed based on evidence-based resource reviews and the input from our multidisciplinary research team. It provides PLWH with reliable symptom management knowledge and personalized symptom management functions. The web-based administration portal also provides the medical staff with convenient functions for case management. Future studies are needed to further test the app’s usability and effectiveness on symptom management. We will refine the SM app based on users’ experience and the collected data.” (Page 10 Line 383-389)